# Microvascular Dysfunction Following Cardioplegic Arrest and Cardiopulmonary Bypass: Impacts of Diabetes and Hypertension

**DOI:** 10.3390/biomedicines13020409

**Published:** 2025-02-07

**Authors:** Meghamsh Kanuparthy, Rishik Manthana, Himanshu Kaushik, Kathy Xiang, Jad Hamze, David Marimekala, Jun Feng, Frank W. Sellke

**Affiliations:** Division of Cardiothoracic Surgery, Department of Surgery, Warren Alpert Medical School, Brown University, Providence, RI 02903, USA; meghamsh_kanuparthy@brown.edu (M.K.); rishik_manthana@brown.edu (R.M.); himanshu_kaushik@brown.edu (H.K.); kathyxiang107@gmail.com (K.X.); jad_hamze@brown.edu (J.H.); david_marimekala@brown.edu (D.M.); jun_feng@brown.edu (J.F.)

**Keywords:** microvascular dysfunction, diabetes, hypertension, cardioplegia, cardiopulmonary bypass

## Abstract

Cardioplegic arrest and cardiopulmonary bypass (CP/CPB) are known to engender microvascular dysfunction in patients undergoing cardiac surgery. These effects are significantly varied by patient comorbidities including diabetes and hypertension. Both diabetes and hypertension are associated with worse outcomes after cardiac surgery, partly related to increased microvascular complications. In this review, we examine several key facets of microvascular dysfunction after CP/CPB: microvascular endothelial and vasomotor dysfunction, altered gene and protein expression, endothelial adherens junction dysfunction, and programmed cell death as they relate to diabetes and hypertension. This review examines both classical techniques, including microvessel reactivity assays, and modern multiomic approaches to characterizing these microvascular changes.

## 1. Introduction

Hypertension (HTN) and diabetes mellitus are well established as significant causes of morbidity and mortality from cardiac disease [1]. Patients with hypertension and diabetes are also significantly at risk for complications when undergoing cardiac surgery. Patients with diabetes have increased rates of mediastinitis, sternal wound infection, and saphenous vein harvest site infections [2]. Diabetes exacerbates postoperative myocardial dysfunction and increases the incidence of low cardiac output syndrome [3]. Additionally, following discharge from the hospital, diabetic patients are more likely to be readmitted, which is especially significant, as diabetic patients have increased morbidity and mortality following surgical revascularization [4,5,6].

Hypertension is similarly a significant risk factor associated with cardiovascular morbidity and mortality and is a clinical challenge for cardiac care teams. Hypertension is associated with a wide array of adverse events such as stroke, heart failure, peripheral arterial disease, dissecting aneurysm, and renal failure [7,8]. After cardiac surgery, coronary artery disease is more prevalent in hypertensive patients as compared to their normotensive counterparts [9]. The resultant coronary atherosclerosis is often more extensive, involves multiple vessels, and progresses more rapidly [10,11]. Hypertension also alters cerebral and renal vascular autoregulation, leading to increased rates of complications in these systems [12,13]. Hypertension remains a significant modifiable risk factor for mortality after coronary artery bypass grafting (CABG) [14].

Cardioplegia and cardiopulmonary bypass (CP/CPB) development has been key in improving the outcomes of cardiac surgery. However, despite advancements in CP/CPB materials and techniques, the widespread inflammatory response and associated organ and vasomotor dysfunction caused by CPB remain a major issue [15]. A particularly important area of concern is the microcirculation, defined as vessels with an internal diameter of less than 200 μm. The microcirculation has been found to be the most significant area of vascular resistance and is a key part of directing tissue perfusion [16]. CP/CPB has been shown to cause microvascular dysfunction in large animal models and patients, leading to impairment in myogenic tone and microvascular response to vasoconstrictive agents. This can present clinically as reduced coronary perfusion leading to myocardial dysfunction as well as systemic hypotension. Examining microvascular dysfunction is an important area of ongoing research. The molecular mechanisms responsible for microvascular constriction and dilation before and after CP/CPB remain unclear given variance in regulatory mechanisms across vascular beds. The intricacies of diabetic and hypertensive regulation pertaining to microvascular function and CP/CPB have been studied by our lab and others. In this review, we discuss recent work highlighting the effects of diabetes and hypertension on vasomotor/endothelial dysfunction, gene and protein expression, vascular permeability, programmed cell death, and cell signaling in the context of CP/CPB.

## 2. Diabetes and Cardioplegic Arrest/Cardiopulmonary Bypass

### 2.1. Microvascular Endothelial and Vasomotor Dysfunction

It is well established in the literature that CP/CPB leads to microvascular endothelial dysfunction across disparate human vascular beds and in various animal models [15,17,18,19]. These in vitro observations of vascular function alterations post-CP/CPB accord with the experience of surgeons and interventionalists as they care for these patients. Those observed macrovascular changes occur in conjunction with and as an extension of microvascular changes which disparately affect different vascular beds across the body [20,21,22]. These alterations are driven by high levels of glucose in the bloodstream causing glycosylation of exposed endothelial elements, driving capillary basement membrane thickening and extracellular matrix proliferation [23]. Further, these glycosylation reactions activate downstream target molecules to induce the production of reactive oxygen species (ROS) to which the damaged endothelium is less able to respond [24]. In combination, these changes significantly alter the vasomotor reactivity to both endogenously and exogenously applied compounds and may explain the resultant vasoplegia after CP/CPB.

Endothelin-1 (ET-1), thromboxane-A-2 (TXA2), and phenylephrine are all vasoconstrictive compounds which act through various receptors; the vasoconstriction induced by all three has been shown to be diminished after CP/CPB, but this was not found to be related to receptor expression [25,26,27]. When patient samples were stratified by diabetes control status, patients with poorly controlled diabetes were found to have diminished vasoconstrictive response as compared to nondiabetic patients [25,26]. Modulation of reactive oxygen species through the use of a mitochondria-specific antioxidant has been shown in diabetic mouse models to ameliorate this discrepancy, further demonstrating the deleterious effects of diabetes-related ROS [28]. The modulation of protein kinase C isoforms has also demonstrated promise in this regard, as the contractile response to phenylephrine was modulated by the differential inhibition or activation of PCK-a [29]. TXA-2-induced coronary arterial constriction occurs through the thromboxane receptor and PLC, but not PKC-*α* [30]. Uncontrolled diabetes has been shown to slow the recovery of endothelial function in coronary and peripheral arterioles following CP/CPB due to increased expression and activation of PKC-a and PKC-b [17,18]. The endothelin (ET) receptors ET-A and ET-B are found in tissues, cells, and vasculature. ET-A receptors promote vasoconstriction and are typically found in the confinement of the coronary microcirculation. ET-B receptors, on the other hand, promote vasodilation and are found in similar areas comparatively less frequently. Activating ET-A receptors and PKC-*α* aids the contractile response to ET-1. However, patients with poorly controlled diabetes demonstrated attenuated contractile responses ET-1 in the peripheral microvasculature in comparison to nondiabetic patients [25,26,31]. Functional studies of small coronary arteries in a porcine model with type 2 diabetes characteristics showed coronary microvascular dysfunction indicated by impaired bradykinin-induced vasodilation due to nitric oxide (NO) loss and reduced vasoconstriction to ET-1, which resulted from decreased ET-A receptor dominance [32].

Similarly, poorly controlled diabetes significantly impairs the relaxation responses of the microvasculature to the endothelium-dependent agents ADP and substance P, compared to well-controlled diabetes or nondiabetic conditions [17,18]. Small (SK) and intermediate (IK) conduction potassium channels are a family of calcium-dependent ion channels associated with vasomotor and neurological function whose pathologic dysfunction has been well documented in states of vascular dysfunction [33]. Impairment of SK_Ca_ and IK_Ca_ channel function in the coronary vasculature is another consequence of CP/CPB [34]. Diabetes also reduces endothelial SK_Ca_/IK_Ca_ currents and hyperpolarization but does not affect the overall gene or protein expression of these channels, suggesting another convergence of influences which worsen CP/CPB-related endothelial dysfunction [35,36,37].

### 2.2. Altered Gene/Protein Expression

In diabetes, hyperglycemia profoundly impacts the myocardium, primarily by promoting endothelial dysfunction. This dysfunction arises through reduced nitric oxide (NO) production and increased pro-inflammatory signaling, which create conditions favorable to atherosclerosis and heightened cardiovascular risk. The decrease in NO, driven by impaired activation of eNOS, leads to vasoconstriction, further compromising cardiovascular health [38]. In addition, the inflammatory response triggered by diabetes, with elevated circulating markers such as C-reactive protein, TNF-α, and interleukin-6, exacerbates vascular damage and increases cardiovascular risk [39]. Furthermore, oxidative stress is heightened due to NADPH consumption, which hampers glutathione (GSH) regeneration. This depletion of GSH reduces nitric oxide production, contributing to vascular dysfunction, atherosclerosis, and impaired myocardial function, thus highlighting the interconnected nature of these processes [24]. Hyperglycemia occurs almost universally during CPB due to factors like catecholamine-induced glucose production, cortisol-mediated insulin resistance, exogenous glucose administration, and CPB-induced hypothermia and poses serious risks for both diabetic and nondiabetic patients undergoing coronary artery bypass grafting [40]. In one study utilizing human atrial tissue samples collected before and after CP/CPB, 851 genes were upregulated in the diabetic group versus 480 in the nondiabetic group, and 48 genes were downregulated in the diabetic cohort as compared to 626 in the nondiabetic cohort. The expressions of 18 genes were upregulated in tandem across groups including key inflammatory transcriptions such as *FOS*, *CYR 61*, and interluekin-6 (IL-6). The pro-apoptotic gene *NR4A1*, stress-responsive gene *DUSP1*, and glucose transporter gene *SLC2A3* also similarly displayed increased expression across these groups. Interestingly, 28 genes were differentially upregulated solely in the diabetic group, including the inflammatory/transcription activators MYC, IL-8, and IL-1, the growth factor VEGF, amphiregulin, and the glucose metabolism-involved gene insulin receptor substrate 1. A summary of these changes is displayed in Figure 1. The epigenome is also significantly affected by the application of CP/CPB, with a response that varies across patient diabetic status. One study which analyzed the methylation of skeletal muscle samples collected from the left internal mammary artery bed during coronary artery bypass grafting demonstrated numerous methylation changes associated with a diabetic state as compared to control patients [41]. Enrichment demonstrated that the single gene pathway most affected in diabetic samples was associated with the Hippo–YAP/TAZ pathway, which a key regulatory pathway associated with cell survival and response to stress [42]. These results suggest tailored myocardial protection and operative strategies for patients with diabetes undergoing cardioplegia/CPB [43].

### 2.3. Downregulation of Endothelial Adherens Junction Proteins

Adherens junctions are specialized cellular structures that facilitate cell–cell adhesion, primarily in endothelial and epithelial tissues by forming a “belt” that wraps around cells [44]. They are composed of cadherins, which are transmembrane proteins that interact with intracellular proteins like β-catenin, plakoglobin, and p120 to connect to the actin cytoskeleton, ensuring structural stability and communication. In vascular endothelium, VE-cadherin is the predominant cadherin and is critical for maintaining endothelial barrier function and vascular permeability [45,46,47,48,49,50]. Protein tyrosine phosphorylation is essential in vasomotor regulation and adjudication of vascular permeability via adherens junctions and endothelial cell contacts [45,51,52]. VE-cadherin phosphorylation is regulated by free radicals and cytokines such as VEGF [53]. Cardioplegia/CPB itself increases phosphorylation of VE-cadherin and decreases β and γ catenins in both pig models and patients undergoing CABG surgery. Poorly controlled diabetes has been shown to downregulate the activation, expression, and localization of endothelial adherens junction proteins in cardioplegia/CPB [45]. Cardioplegia/CPB disrupts endothelial cadherin structure in the coronary endothelium, particularly in diabetic vessels [51,52]. Increased tyrosine phosphorylation and VE-cadherin degradation weakens cell–cell junctions, leading to greater vascular permeability and endothelial dysfunction. These changes may worsen outcomes in diabetic patients after cardiac surgery.

### 2.4. Increased Programmed Cell Death

Cardioplegia/CPB is currently associated with programmed cell death, such as apoptosis [54,55,56]. Cardioplegia/CPB can induce programmed cell death and survival signaling through the caspase-dependent and intrinsic pathways in myocardial and endothelial cells. Uncontrolled diabetes is typically associated with increased myocardial apoptosis and expression of apoptosis mediators, also present during cardioplegia/CPB [57]. Diabetic myocardium demonstrated reduction of the cardioprotective STAT3 pathways after cardioplegia/CPB and cardiac surgery in comparison to nondiabetic myocardium [58]. Lower baseline urocortin levels in diabetic hearts that fail to increase after cardioplegic arrest are associated with increased apoptosis and postsurgical cardiac dysfunction [59]. Cardiac surgery was associated with a significant depletion of LC2-I, LC3-II, beclin-1, and autophagy 5-12 as well as changes in the flux marker p62, indicative of autophagic flux. Autophagy, or the removal of dysfunctional organic components, is thought to be a mechanism of protection against cardiac damage [60]. Cross-clamp time during surgery was directly correlated with p62 changes and had an inverse correlation with mortality and morbidity. Operation-related ischemia is associated with significant changes in expression in 14 of 94 autophagy-related genes (ATGs). In particular, key autophagy machinery components, including ATG4A, ATG4C, and ATG4D, were upregulated. Chaperone-mediated autophagy was also elevated, as indicated by increased levels of the heat shock proteins HSPA8 and HSP90AA1, as well as α-synuclein. Additionally, there was an upregulation of tumor necrosis factor-related apoptosis-inducing ligand (TRAIL), MAPK8, and BCL2L1. Autophagy activation was confirmed by higher LC3-I levels and an increased LC3-II/LC3-I ratio, a common method to quantify autophagic activity in cells [60].

### 2.5. Effect of Select Antidiabetic Medications on Microvascular Reactivity

Antidiabetic medications, particularly glucagon-like peptide-1 (GLP-1) receptor agonists and dipeptidyl peptidase-4 (DPP-4) inhibitors, have been shown to improve microvascular dysfunction through modulation of insulin signaling, reduction of oxidative stress, and enhancement of endothelial function, which are critical for maintaining microvascular integrity. SGLT2 inhibitors have been found to normalize Ca-dependent signaling in the setting of ischemia and reperfusion in both in vitro and murine models [61,62]. Antidiabetic medications enhance insulin sensitivity, which is crucial for microvascular function. GLP-1 receptor agonists, such as semaglutide, increase the bioavailability of insulin by promoting its secretion from pancreatic β-cells while simultaneously suppressing glucagon release [63,64]. This dual action helps to lower blood glucose levels and improve insulin-mediated capillary recruitment. Improved insulin signaling can lead to enhanced microvascular reactivity, and insulin itself can directly induce relaxation in resistance vessels [65]. Hyperglycemia, a hallmark of diabetes, leads to increased production of reactive oxygen species (ROS), contributing to oxidative stress and endothelial dysfunction [66]. Antidiabetic medications, particularly metformin and GLP-1 receptor agonists, have been shown to reduce oxidative stress by enhancing mitochondrial function and promoting the expression of antioxidant enzymes. For example, metformin activates the AMP-activated protein kinase (AMPK) pathway, which plays a crucial role in cellular energy homeostasis and has been linked to reduced oxidative stress in endothelial cells [67]. GLP-1 receptor agonists have also been shown to enhance nitric oxide (NO) availability through this same pathway, which may help combat oxidative stresses associated with CP/CPB [68]. Extensive clinical evidence has suggested that GLP1 agonists carry cardioprotective benefits, yet studies specifically in patients who have undergone cardiac surgery and thus been exposed to CP/CPB have yet to identify a clear clinical benefit in the short term and may require longer-term follow-up [69,70].

## 3. Hypertension and Cardioplegic Arrest/Cardiopulmonary Bypass

### 3.1. Microvascular Endothelial and Vasomotor Dysfunction

Hypertension, like diabetes, is a disease that directly affects the endothelium both before and after the initiation of cardiopulmonary bypass. This endothelial dysfunction has been associated with up to a fourfold increase in adverse cardiovascular outcomes in patients with worsened response to endothelial dependent vasodilators [71]. Endothelial-derived relaxing factor, also known as nitric oxide (NO), is produced from L-arginine and increases cyclic guanosine monophosphate levels in the cells, leading to a transient decrease in intracellular calcium levels and vascular smooth muscle relaxation [72]. Hypertensive states have been shown to induce dysregulation of NO production through the production of increased oxidative stress [73]. Beyond its function as a vasodilator, NO inhibits platelet aggregation, the attachment and translocation of neutrophils to vessel walls, and the proliferation of smooth muscle, which all in turn have an antiatherosclerotic influence on microvasculature [74,75,76,77]. In patients with hypertension, this NO dysregulation is driven by suppression of the expression and activity of endothelial nitric oxide synthase (eNOS) by increasing reactive oxygen species proliferation. These ROS species deplete the cellular environment of L-arginine and of tetrahydrobiopterin, a key cofactor of eNOS [78,79,80,81]. Ischemia and reoxygenation during CP/CPB are associated with the generation of large amounts of free radicals via neutrophil NADPH peroxisome activation while passing through the extracorporeal oxygenation circuit, increased superoxide production, and increased neutrophil elastase production [82,83,84]. In vitro studies have shown that these effects work synergistically to impair baseline microvascular function after CP/CPB. One study utilizing samples collected from right atrial venous cannulation sites from patients undergoing on-pump cardiac surgery demonstrated that microvessels collected from patients with poorly controlled hypertension exhibited significantly greater myogenic tone as compared to patients with well-controlled hypertension or non-hypertensive patients [85].

Beyond endogenous NO production, hypertension modulates the vascular endothelium’s response to various vasoactive compounds after exposure to cardiopulmonary bypass [86]. Microvascular reactivity studies serve as an important methodological instrument for functionally assessing vasomotor function and generally involve the careful dissection of microvessels, cannulation, pressurization, and measurement of pressure or diameter in response to vasoactive substances instilled into circulating baths [87]. One crucial modulator of vasomotor tone is 5-hydroxytryptamine (5-HT), more commonly known as serotonin, which can serve either a vasodilatory function, primarily through the 5HT2B and 5HT7 receptors, or a vasoconstrictive function, principally through 5HT2A receptors [88]. In patients with uncontrolled hypertension, cardiopulmonary bypass was found to induce increased contraction of coronary microvasculature in response to 5-HT, which may be driven by an associated decrease in the expression of the 5HT1A receptor in post-CP/CPB tissue [89]. Thromboxane A2 (TXA2) is another major endothelial-derived vasoconstrictor that is derived from arachidonic acid. Microvascular response to TXA2 through the TXA2R receptor is known to be downregulated after cardiopulmonary bypass [30,90]. Dose-dependent microvascular contractile response to U46619, an analog of TXA2, has been shown to be significantly increased in tissue from patients with poorly controlled hypertension as compared to those with well-controlled hypertension or non-hypertensive patients. The response to exogenous vasoactive chemicals is also modulated in similar ways. Phenylephrine, a vasoconstrictor commonly used in the perioperative care of patients which acts through the alpha-1 adrenergic receptor, similarly demonstrated increased contractile response in those patients with poorly controlled hypertension [85,91]. Conversely, microvascular response to endothelium-dependent vasodilators appears attenuated. In yet-to-be-published work from the laboratory of Drs. Frank Sellke and Jun Feng, a diminished vasomotor response to bradykinin, a coronary vasodilator which acts through the B2 receptor, and to adenosine diphosphate (ADP) has been established. A summary of these findings is shown in Figure 2.

### 3.2. Altered Gene/Protein Expression

By exacerbating endothelial injury, hypertension activates protein kinase C pathways and disrupts the production of vasodilators such as nitric oxide. With the increased inflammation from protein kinase C pathways and the lack of nitric oxide, it becomes substantially more difficult for microvessels to properly control their diameter and regulate blood flow [89]. Hypertension-induced vascular dysfunction and vasoconstriction are also significantly influenced by gene expression changes involving angiotensin II (Ang II) and endothelin-1 (ET-1). Ang II upregulates endothelin receptor A (ETAR) expression in vascular smooth muscle cells and enhances ET-1 binding, leading to heightened vasoconstriction. This effect is mediated through protein kinase C and extracellular signal-regulated kinase (ERK) signaling pathways, illustrating the critical interplay between Ang II and ET-1 in elevating blood pressure [92]. Additionally, hypertension is associated with an imbalance in matrix metalloproteinase (MMP) activity, leading to extracellular matrix (ECM) degradation and disrupted vascular remodeling associated with hypertension-related vascular disease. Therapeutic approaches targeting Ang II, ETAR, and MMP activity could help mitigate hypertension’s impact on vascular health [93]. Hypertension is a major risk factor that exacerbates coronary microvascular dysfunction during CP/CPB, with studies showing increased vasoconstrictive responses in hypertensive patients and shifts in serotonin receptor expression that heighten vascular reactivity [89].

### 3.3. Downregulation of Endothelial Adherens Junction Proteins

Adherens junctions play a key role in the regulation of endothelial cell function by modulating their ability to connect to other nearby cells. In their functional state, cadherins, particularly VE-cadherin, the primary cadherin in vascular tissue, act to form a dimerized bond which anchors cells together and to their actin cytoskeleton [94,95]. This interaction is not only crucial for regulating tissue integrity and preventing vasogenic edema but is also critical for the mechanical resistance to stress and functions as part of the VE-cadherin mechanosensory complex [96,97]. Hypertension-related increases in vascular shear stress have been associated with decreases in the junctional proteins VE-cadherin and β-catenin [98]. There are several mechanisms that may explain the downregulation of these proteins. Hypertension often results in increased oxidative stress and ROS production, which can cause significant damage to adherens junctions [45]. CP/CPB also generates significant ROS and is known to increase VE-cadherin phosphorylation and degradation through the action of SRC kinase [52,99]. By increasing shear stress and inflammation via cytokines, hypertension upregulates the phosphorylation of junctional proteins like VE-cadherin, resulting in their internalization and degradation [45]. Hypertension can also increase the expression of matrix metalloproteinases (MMPs) that break down and destabilize endothelial adherens junctions. These adherens junctions are vital for maintaining tissue hydration and preventing excessive fluid and protein leakage, and their downregulation can further exacerbate other cardiovascular conditions [98].

### 3.4. Effect of Select Antihypertensive Medications on Microvascular Reactivity

Antihypertensive agents, including beta-blockers, calcium channel blockers (CCBs), angiotensin-converting enzyme inhibitors (ACEIs), and angiotensin receptor blockers (ARBs), all have significant impacts on the microvasculature, but there is limited evidence in the setting of CP/CPB. However, there is significant evidence from cardiology literature which utilizes angiographic methods to identify how these agents target endothelial dysfunction and reduce myocardial oxygen demand, thereby enhancing coronary blood flow (CBF) and coronary flow reserve (CFR) [100]. Beta-blockers, particularly third-generation agents like nebivolol and carvedilol, provide significant benefits by improving endothelial function through mechanisms such as nitric oxide (NO) release, reduced oxidative stress, and improved diastolic perfusion [101,102]. Clinical trials have shown that beta-blockers effectively reduce ischemic episodes and enhance exercise tolerance due to these mechanisms [103].

CCBs also contribute to microvascular modulation by reducing myocardial oxygen demand and relaxing microvascular tone [104]. In vivo use of CCBs in reducing endothelial damage as an agent in CPB circuits have shown limited efficacy, but their vasodilatory effects and ability to mitigate free radical injuries may support long term endothelial health [105]. Evidence suggests that long-acting L-type CCBs are more effective than short-acting agents, particularly when combined with statins [106]. Statins enhance endothelial function through increased NO bioavailability, antioxidant effects, and anti-inflammatory properties, demonstrating synergistic benefits with CCBs in improving CFR and exercise-induced ischemia [107]. Similarly, ACEIs and ARBs improve microvascular dysfunction by reducing reactive oxygen species and stimulating NO production [108]. ACEIs have shown superior effects on endothelial-dependent vasodilation and CFR, making them highly effective in hypertensive patients with angina [109]. Further studies evaluating these drugs’ impact in the setting of CP/CPB are needed.

## 4. Challenges and Future Directions

As cardiac surgery continues to advance, several challenges and opportunities for innovation have emerged, which are critical to improving patient outcomes and optimizing resource utilization. Future work should focus on developing personalized approaches to cardioplegia tailored to individual patient comorbidities, leveraging metabolic profiles and preexisting conditions to refine myocardial protection strategies during surgery.

Additionally, emerging pharmacological interventions, such as GLP-1 receptor agonists, SGLT1/2 inhibitors, and DPP-4 inhibitors, offer promise not only for diabetes management but also for preventing cardiac remodeling and warrant further investigation for integration into perioperative care. The complex interplay between obesity, hypertension, and diabetes highlights the need for optimized management of these interconnected conditions through lifestyle interventions and pharmacotherapy. Moreover, with the rising prevalence of diabetes, particularly in resource-limited settings, understanding genetic variability and its influence on disease progression and treatment response is essential to developing effective and accessible strategies for diverse populations.

The application of multiomic technologies, including genomics, transcriptomics, proteomics, and metabolomics, provides an unprecedented opportunity to comprehensively investigate the molecular changes associated with cardiopulmonary bypass and related comorbidities, guiding targeted interventions. For example, molecular techniques have identified a specific mutation of Philadelphia chromosome negative myeloproliferative disease that may be associated with endothelial dysfunction [110]. These techniques will allow for specific changes to CP/CPB systems to account for these changes.

Lastly, addressing the significant variability in surgical practices across hospitals will be crucial for translating innovations into widespread clinical application, requiring the standardization of protocols, enhanced inter-institutional collaboration, and robust evaluation mechanisms in diverse healthcare settings. By tackling these challenges and leveraging emerging opportunities, the field of cardiac surgery can continue to evolve, driving improvements in care for patients worldwide.

### Interactions with COVID-19

The COVID-19 pandemic has highlighted the complex interplay between diabetes, hypertension, and microvascular dysfunction, particularly in the context of adverse outcomes associated with SARS-CoV-2 infection [111]. Both diabetes and hypertension are prevalent comorbidities that significantly affect the clinical course of COVID-19, leading to increased morbidity and mortality. In patients with diabetes, chronic hyperglycemia leads to the formation of advanced glycation end-products (AGEs), which can activate inflammatory pathways and promote oxidative stress [112]. This oxidative stress further damages endothelial cells, exacerbating microvascular dysfunction. In hypertensive patients, increased levels of angiotensin II can lead to endothelial cell injury and promote vasoconstriction, contributing to impaired microvascular reactivity [113]. COVID-19 infection triggers a hyper-inflammatory response, often referred to as a “cytokine storm,” which can be particularly detrimental in patients with pre-existing conditions like diabetes and hypertension. Elevated levels of pro-inflammatory cytokines, such as interleukin-6 (IL-6) and tumor necrosis factor-alpha (TNF-α), have been associated with severe COVID-19 outcomes [114].

In patients with diabetes and hypertension, the risk of thrombotic events after cardiac surgery is heightened due to underlying endothelial dysfunction and altered coagulation profiles [115]. Patients with a history of COVID-19 infection have been found to have higher odds of developing complications such as deep vein thrombosis (DVT), pulmonary embolism (PE), and myocardial injury after cardiac surgery [116,117]. These findings underscore the importance of thorough preoperative assessments and potential interventions to address these risks.

## 5. Conclusions

Diabetes and hypertension are both associated with vascular dysfunction across various tissues, including the microvasculature. A better understanding of the mechanisms regulating microvascular tone during and after cardiac surgery may inform the development of strategies to mitigate the detrimental effects of cardioplegia and cardiopulmonary bypass (CPB). Given that cardioplegia and CPB are integral to most cardiac surgeries—and that patients with diabetes or hypertension experience higher rates of postoperative complications and mortality—future clinical efforts should focus on preserving microvascular integrity and reducing the adverse impact of extracorporeal circulation on vasomotor regulation and organ function.

## Figures and Tables

**Figure 1 biomedicines-13-00409-f001:**
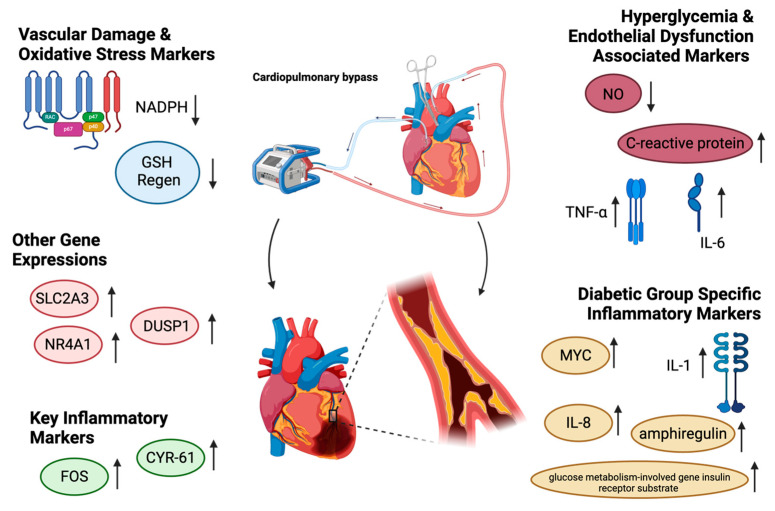
Selected gene and protein expressional changes associated with CP/CPB in the setting of diabetes.

**Figure 2 biomedicines-13-00409-f002:**
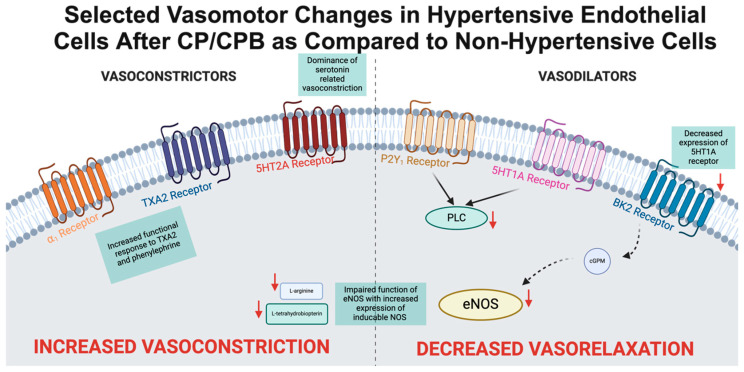
Selected vasomotor changes in hypertensive endothelial cells after CP/CPB as compared to non-hypertensive cells.

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
