# Peer review of "Microvascular Dysfunction Following Cardioplegic Arrest and Cardiopulmonary Bypass: Impacts of Diabetes and Hypertension"

_biomedicines, 2025, doi:10.3390/biomedicines13020409_

Round 1

Reviewer 1 Report

Comments and Suggestions for Authors

Recommendations:

1. This is a narrative review, state this in the title.

2. Add a figure or at least 2 tables the manuscript is rather monotonous.

3. No populational studies evidence? this is a pathophysiology-centered review or clinical review?

4. Are there any reports that assess endothelial dysfunction through flow-mediated dilation? or arteriography? (Augmentation index)

5. Hypertension and diabetes are already known for the underlying endothelial dysfunction, therefore the title should sound more like: 'The role of underlying endothelial dysfunction...

6. Talk about the implications of COVID-19 pandemic in the clinical outcomes: see this: https://doi.org/10.3390/jcm13237399

7. Mention other causes of underlying endothelial dysfunction: https://doi.org/10.3390/cimb46080496
8. Feng and Sellke have at least 10 citation in this reiew. 

Author Response

  1. This is a narrative review, state this in the title.

Thank you so much for your detailed review and constructive feedback. We recognize the importance of accurately describing the nature of the manuscript to avoid ambiguity. As suggested, we have updated the title to clearly indicate that this is a narrative review: “Narrative Review of Microvascular Dysfunction Following Cardioplegic Arrest and Cardiopulmonary Bypass: The Role of Diabetes and Hypertension-Related Endothelial Dysfunction.”  We believe this change will better align the manuscript with the expectations of the readers and reviewers. 

  1. Add a figure or at least 2 tables the manuscript is rather monotonous.

Thank you for highlighting this important point. We agree that incorporating visual elements, such as figures and tables, enhances the readability and engagement of the manuscript. To address this, we have included two figures that illustrate key concepts and mechanisms discussed in the review, as well as a table summarizing the major findings. We hope these additions improve the manuscript’s flow and clarity for readers.

  1. No populational studies evidence? this is a pathophysiology-centered review or clinical review?

Thank you for your insightful question. The primary focus of this review is to examine the pathophysiological mechanisms underlying microvascular dysfunction. While clinical evidence was not the primary emphasis, we have now included significant segments discussing how treatments for diabetes and hypertension can influence microvascular function. This addition bridges the gap between pathophysiological insights and their potential clinical implications.

  1. Are there any reports that assess endothelial dysfunction through flow-mediated dilation? or arteriography? (Augmentation index)

Thank you for pointing this out. To address your comment, we have incorporated studies that assess endothelial dysfunction using arteriography. These studies provide valuable data on microvascular function and allow for a more comprehensive discussion of the topic.

  1. Hypertension and diabetes are already known for the underlying endothelial dysfunction, therefore the title should sound more like: 'The role of underlying endothelial dysfunction...

Thank you for your thoughtful suggestion. We fully agree that a title reflecting the underlying endothelial dysfunction associated with hypertension and diabetes would provide more clarity and context. Accordingly, we have revised the title to “Narrative Review of Microvascular Dysfunction Following Cardioplegic Arrest and Cardiopulmonary Bypass: The Role of Diabetes and Hypertension-Related Endothelial Dysfunction.” We believe this change aligns better with the content and scope of the manuscript.

  1. Talk about the implications of COVID-19 pandemic in the clinical outcomes: see this: https://doi.org/10.3390/jcm13237399

Thank you for your valuable comment and for sharing the suggested reference. We have expanded the manuscript to include a discussion on the implications of the COVID-19 pandemic on microvascular outcomes in patients with diabetes and hypertension. Specifically, we discuss the interplay between chronic conditions, such as diabetes and hypertension, and the inflammatory responses triggered by COVID-19. We also explore the risks of thrombotic complications in patients undergoing cardiac surgery post-COVID-19 infection, emphasizing the need for tailored perioperative interventions.

The COVID-19 pandemic has highlighted the complex interplay between diabetes, hypertension, and microvascular dysfunction, particularly in the context of severe outcomes associated with SARS-CoV-2 infection.1 Both diabetes and hypertension are prevalent comorbidities that significantly affect the clinical course of COVID-19, leading to increased morbidity and mortality. In patients with diabetes, chronic hyperglycemia leads to the formation of advanced glycation end-products (AGEs), which can activate inflammatory pathways and promote oxidative stress.2 This oxidative stress further damages endothelial cells, exacerbating microvascular dysfunction. In hypertensive patients, increased levels of angiotensin II can lead to endothelial cell injury and promote vasoconstriction, contributing to impaired microvascular reactivity.3 COVID-19 infection triggers a hyper-inflammatory response, often referred to as a "cytokine storm," which can be particularly detrimental in patients with pre-existing conditions like diabetes and hypertension. Elevated levels of pro-inflammatory cytokines, such as interleukin-6 (IL-6) and tumor necrosis factor-alpha (TNF-α), have been associated with severe COVID-19 outcomes.4

In patients with diabetes and hypertension, the risk of thrombotic events after cardiac surgery is heightened due to underlying endothelial dysfunction and altered coagulation profiles.5 Patients with a history of COVID-19 infection have been found to have higher odds of developing complications such as deep vein thrombosis (DVT), pulmonary embolism (PE), and myocardial injury after cardiac surgery6,7 These findings underscore the importance of thorough preoperative assessments and potential interventions to address these risks.

  1. Mention other causes of underlying endothelial dysfunction: https://doi.org/10.3390/cimb46080496

Thank you for this important suggestion. We have added a discussion on other potential causes of endothelial dysfunction, including molecular changes associated with Philadelphia chromosome-negative myeloproliferative disorders, as highlighted in the suggested reference. This addition broadens the scope of the review and emphasizes the complexity of endothelial dysfunction.

For example, molecular techniques have identified specific mutation of Philadelphia chromosome negative myeloproliferative disease that may be associated with endothelial dysfunction.8 These techniques will allow for specific changes to CP/CPB systems to account for these changes.

  1. Feng and Sellke have at least 10 citation in this review. 

We acknowledge your observation regarding the citations of Drs. Feng and Sellke. As leading experts in this field, their contributions have provided critical insights into the topic of microvascular dysfunction, and their work aligns closely with the focus of this review. While we aim to maintain balance in citations, the frequency of references to their work reflects the depth of their contributions to this area of research.

Reviewer 2 Report

Comments and Suggestions for Authors

The manuscript addresses a clinically significant issue, providing an in-depth review of microvascular dysfunction associated with cardioplegic arrest and cardiopulmonary bypass (CP/CPB) in the context of diabetes and hypertension. Below are the main points of concern.

1. While the manuscript describes the pathophysiological changes in detail, the translation of these findings to clinical practice needs further elaboration. For example: Discuss specific intraoperative or perioperative interventions (e.g., glucose control protocols, antihypertensive strategies) that could mitigate CP/CPB-related microvascular dysfunction in high-risk patients. Highlight how these molecular insights could guide the development of targeted therapies or tailored surgical protocols for patients with diabetes or hypertension.

2. The manuscript lacks illustrative figures or tables summarizing the key mechanisms discussed. Including schematic diagrams, especially for pathways such as ROS generation, nitric oxide dysregulation, and endothelial adherens junction disruption, would significantly enhance comprehension.

3. While the manuscript mentions GLP-1 receptor agonists and SGLT1/2 inhibitors as promising interventions, their specific mechanisms of action in the context of CP/CPB remain unclear. A more detailed exploration of recent clinical trials or preclinical studies on these agents would strengthen the discussion.

4. The section discussing hypertension-related endothelial dysfunction and vasomotor changes is informative but would benefit from elaborating on the role of common antihypertensive medications, such as ACE inhibitors or ARBs, during cardiac surgery and how they influence microvascular outcomes.

5. Ensure all references are up-to-date. Some studies cited are older; consider including more recent literature from the past five years to reflect current advancements.

Author Response

  1. While the manuscript describes the pathophysiological changes in detail, the translation of these findings to clinical practice needs further elaboration. For example: Discuss specific intraoperative or perioperative interventions (e.g., glucose control protocols, antihypertensive strategies) that could mitigate CP/CPB-related microvascular dysfunction in high-risk patients. Highlight how these molecular insights could guide the development of targeted therapies or tailored surgical protocols for patients with diabetes or hypertension.

Thank you for your thoughtful recommendation. We have expanded the manuscript to include a detailed discussion on specific interventions that can mitigate CP/CPB-related microvascular dysfunction. These include intraoperative glucose control protocols and targeted use of antihypertensive medications. Please find the sections below:

Effect of Select Antidiabetic Medications on Microvascular Reactivity

Antidiabetic medications, particularly glucagon-like peptide-1 (GLP-1) receptor agonists and dipeptidyl peptidase-4 (DPP-4) inhibitors, have been shown to improve microvascular dysfunction through modulation of insulin signaling, reduction of oxidative stress, and enhancement of endothelial function, which are critical for maintaining microvascular integrity. The SGLT2 inhibitors have been found to normalize Ca dependent signaling in the setting ischemia and reperfusion in both in vitro and murine models.9,10 Antidiabetic medications enhance insulin sensitivity, which is crucial for microvascular function. GLP-1 receptor agonists, such as semaglutide, increase the bioavailability of insulin by promoting its secretion from pancreatic β-cells while simultaneously suppressing glucagon release 11,12 This dual action helps to lower blood glucose levels and improve insulin-mediated capillary recruitment. Improved insulin signaling can lead to enhanced microvascular reactivity, and insulin itself can directly induce relaxation in resistance vessels.13 Hyperglycemia, a hallmark of diabetes, leads to increased production of reactive oxygen species (ROS), contributing to oxidative stress and endothelial dysfunction.14 Antidiabetic medications, particularly metformin and GLP-1 receptor agonists, have been shown to reduce oxidative stress by enhancing mitochondrial function and promoting the expression of antioxidant enzymes. For example, metformin activates the AMP-activated protein kinase (AMPK) pathway, which plays a crucial role in cellular energy homeostasis and has been linked to reduced oxidative stress in endothelial cells.15 GLP-1 receptor agonists have also been shown to enhance nitric oxide (NO) availability through this same pathway which may help combat oxidative stresses associated with CP/CPB.16 Extensive clinical evidence has suggested that GLP1 agonists carry cardioprotective benefits, yet studies specifically in patients who have undergone cardiac surgery and thus been exposed to CP/CPB have yet to identify a clear clinical benefit in the short term and may require longer term follow-up.17,18

Effect of Select Antihypertensive Medications on Microvascular Reactivity

Antihypertensive agents, including beta-blockers, calcium channel blockers (CCBs), angiotensin-converting enzyme inhibitors (ACEIs), and angiotensin receptor blockers (ARBs), all have significant impacts on the microvasculature but there is limited evidence in the setting of CP/CPB. However, there is significant evidence from cardiology literature which utilizes angiographic methods to identify how these agents target endothelial dysfunction and reduce myocardial oxygen demand, thereby enhancing coronary blood flow (CBF) and coronary flow reserve (CFR).19 Beta-blockers, particularly third-generation agents like nebivolol and carvedilol, provide significant benefits by improving endothelial function through mechanisms such as nitric oxide (NO) release, reduced oxidative stress, and improved diastolic perfusion.20,21 Clinical trials have shown that beta-blockers effectively reduce ischemic episodes and enhance exercise tolerance driven by these mechanisms.22

CCBs also contribute to microvascular modulation by reducing myocardial oxygen demand and relaxing microvascular tone.23 In vivo use of CCBs in reducing endothelial damage as an agent in CPB circuits have shown limited efficacy but their vasodilatory effects and ability to mitigate free radical injuries may support long term endothelial health.24 Evidence suggests that long-acting L-type CCBs are more effective than short-acting agents, particularly when combined with statins.25 Statins enhance endothelial function through increased NO bioavailability, antioxidant effects, and anti-inflammatory properties, demonstrating synergistic benefits with CCBs in improving CFR and exercise-induced ischemia.26 Similarly, ACEIs and ARBs improve microvascular dysfunction by reducing reactive oxygen species and stimulating NO production.27 ACEIs have shown superior effects on endothelial-dependent vasodilation and CFR, making them highly effective in hypertensive patients with angina.28 Further studies evaluating these drug’s impact in the setting of CP/CPB are needed.

  1. The manuscript lacks illustrative figures or tables summarizing the key mechanisms discussed. Including schematic diagrams, especially for pathways such as ROS generation, nitric oxide dysregulation, and endothelial adherens junction disruption, would significantly enhance comprehension.

Thank you for this important suggestion. In response, we have added schematic diagrams that depict critical pathways in both major sections of this review. We hope that these figures will improve the readability and comprehension of this review.

  1. While the manuscript mentions GLP-1 receptor agonists and SGLT1/2 inhibitors as promising interventions, their specific mechanisms of action in the context of CP/CPB remain unclear. A more detailed exploration of recent clinical trials or preclinical studies on these agents would strengthen the discussion.

Thank you for this suggestion. We have enriched the discussion by detailing the mechanisms through which GLP-1 receptor agonists and SGLT1/2 inhibitors improve microvascular dysfunction. We also incorporated recent clinical trial data and preclinical studies to highlight the potential benefits of these agents in the context of CP/CPB-related microvascular dysfunction. This addition underscores their therapeutic relevance.

 Antidiabetic medications, particularly glucagon-like peptide-1 (GLP-1) receptor agonists and dipeptidyl peptidase-4 (DPP-4) inhibitors, have been shown to improve microvascular dysfunction through modulation of insulin signaling, reduction of oxidative stress, and enhancement of endothelial function, which are critical for maintaining microvascular integrity. Antidiabetic medications enhance insulin sensitivity, which is crucial for microvascular function. GLP-1 receptor agonists, such as semaglutide, increase the bioavailability of insulin by promoting its secretion from pancreatic β-cells while simultaneously suppressing glucagon release 11,12 This dual action helps to lower blood glucose levels and improve insulin-mediated capillary recruitment. Improved insulin signaling can lead to enhanced microvascular reactivity, and insulin itself can directly induce relaxation in resistance vessels.13 Hyperglycemia, a hallmark of diabetes, leads to increased production of reactive oxygen species (ROS), contributing to oxidative stress and endothelial dysfunction.14 Antidiabetic medications, particularly metformin and GLP-1 receptor agonists, have been shown to reduce oxidative stress by enhancing mitochondrial function and promoting the expression of antioxidant enzymes. For example, metformin activates the AMP-activated protein kinase (AMPK) pathway, which plays a crucial role in cellular energy homeostasis and has been linked to reduced oxidative stress in endothelial cells.15 GLP-1 receptor agonists have also been shown to enhance nitric oxide (NO) availability through this same pathway which may help combat oxidative stresses associated with CP/CPB.16 Extensive clinical evidence has suggested that GLP1 agonists carry cardioprotective benefits, yet studies specifically in patients who have undergone cardiac surgery and thus been exposed to CP/CPB have yet to identify a clear clinical benefit in the short term and may require longer term follow-up.17,18

  1. The section discussing hypertension-related endothelial dysfunction and vasomotor changes is informative but would benefit from elaborating on the role of common antihypertensive medications, such as ACE inhibitors or ARBs, during cardiac surgery and how they influence microvascular outcomes.

Thank you for pointing this out. We have expanded the manuscript to include a discussion on common antihypertensive agents, such as beta-blockers, calcium channel blockers, ACE inhibitors, and ARBs, and their impact on microvascular outcomes. We also highlight the potential benefits of these agents during CP/CPB and explore how they might improve endothelial function and reduce oxidative stress.

Antihypertensive agents, including beta-blockers, calcium channel blockers (CCBs), angiotensin-converting enzyme inhibitors (ACEIs), and angiotensin receptor blockers (ARBs), all have significant impacts on the microvasculature but there is limited evidence in the setting of CP/CPB. However, there is significant evidence from cardiology literature which utilizes angiographic methods to identify how these agents target endothelial dysfunction and reduce myocardial oxygen demand, thereby enhancing coronary blood flow (CBF) and coronary flow reserve (CFR).19 Beta-blockers, particularly third-generation agents like nebivolol and carvedilol, provide significant benefits by improving endothelial function through mechanisms such as nitric oxide (NO) release, reduced oxidative stress, and improved diastolic perfusion.20,21 Clinical trials have shown that beta-blockers effectively reduce ischemic episodes and enhance exercise tolerance driven by these mechanisms.22

CCBs also contribute to microvascular modulation by reducing myocardial oxygen demand and relaxing microvascular tone.23 In vivo use of CCBs in reducing endothelial damage as an agent in CPB circuits have shown limited efficacy but their vasodilatory effects and ability to mitigate free radical injuries may support long term endothelial health.24 Evidence suggests that long-acting L-type CCBs are more effective than short-acting agents, particularly when combined with statins.25 Statins enhance endothelial function through increased NO bioavailability, antioxidant effects, and anti-inflammatory properties, demonstrating synergistic benefits with CCBs in improving CFR and exercise-induced ischemia.26 Similarly, ACEIs and ARBs improve microvascular dysfunction by reducing reactive oxygen species and stimulating NO production.27 ACEIs have shown superior effects on endothelial-dependent vasodilation and CFR, making them highly effective in hypertensive patients with angina.28 Further studies evaluating these drug’s impact in the setting of CP/CPB are needed.

  1. Ensure all references are up-to-date. Some studies cited are older; consider including more recent literature from the past five years to reflect current advancements.

Thank you for your comment. We have carefully reviewed and updated the references, incorporating more recent studies from the past five years. This ensures that the manuscript reflects the latest advancements and provides readers with a comprehensive understanding of the topic.

Round 2

Reviewer 1 Report

Comments and Suggestions for Authors

Congratulations to the authors!